# Effects of Low-Temperature Tempering on Microstructure and Properties of the Laser-Cladded AISI 420 Martensitic Stainless Steel Coating

**Hongmei Zhu, Yongzuo Li, Baichun Li, Zhenyuan Zhang and Changjun Qiu ***

School of Mechanical Engineering, University of South China, Hengyang 421001, China;
meizihong999@126.com (H.Z.); liyongzuo106@126.com (Y.L.); libaichun106@126.com (B.L.);
zhangzhenyuan106@126.com (Z.Z.)
* Correspondence: 420005374203@usc.edu.cn; Tel./Fax: +86-734-828-1661

**Abstract:** Post-treatment is crucial to improve the comprehensive performance of laser-cladded martensitic stainless steel coatings. In this work, a low-temperature tempering treatment (210 °C), for the first time, was performed on the laser-cladded AISI 420 martensitic stainless steel coating. The microstructure and properties of the pre- and post-tempering specimens were carefully investigated by XRD, SEM, TEM, a micro-hardness tester, a universal material testing machine and an electrochemical workstation. The results show that the as-cladded AISI 420 stainless steel coating mainly consisted of martensite, austenite, $Fe_3C$ and $M_{23}C_6$ carbides. The phase constituent of the coating remained the same, however, the martensite decomposed into finer tempered martensite with the precipitation of numerous nano-sized $Fe_3C$ carbides and reverted austenite in the as-tempered specimen. Moreover, a slight reduction was found in the micro-hardness and tensile strength, while a significant increase in elongation was achieved after tempering. The fractography showed a transition from brittle fracture to ductile fracture accordingly. The as-tempered coating exhibited a striking combination of mechanical properties and corrosion resistance. This work can provide a potential strategy to enhance the overall properties of the laser-deposited Fe-based coating for industrial applications.

**Keywords:** laser cladding; low-temperature tempering; stainless steel coating; microstructure; properties

## 1. Introduction

Laser cladding, as a simple but efficient surface modification technique, has been extensively utilized in different industrial fields such as surface coating functionalization, component repair and additive manufacturing under harsh service conditions [1,2]. AISI 420 martensitic stainless steel (SS) has been considered as one of the potential alloys for creating additive manufacturing functional coatings/components owing to its high mechanical properties, moderate corrosion resistance and tailored properties by the subsequent heat treatment [2–4].

However, there was only a little reporting in the literature on laser-cladded AISI 420 SS coatings [3–7]. For instance, Krakhmalev et al. [3] investigated the thermal cycling by numerical simulation and the in-situ microstructural evolution by an experiment with AISI 420 SS prepared by a selective laser melting (SLM) technique. Zhang et al. [4] studied the heat transfer process of the laser-cladded AISI 420 with 4% Mo, and the erosion and corrosion resistance of the laser-cladded AISI 420/VC metal matrix composites [5]. In order to improve the properties of the laser-cladded AISI 420 SS coating, Alam et al. [6] explored the effects of the post-cladding heat treatment (PCHT, 565 °C for 1 h) on the micro-hardness variation and residual stress of laser-cladded specimens. They

found that PCHT could lead to not only reductions of micro-hardness and residual stress, but also the precipitation of tempered martensite and more carbides [6]. Recently, Sun et al. [7] reported an in-situ quench and tempering treatment by introducing a laser idle time of 80 s between clad tracks, and demonstrated a significant improvement in the ductility of the AISI 420 SS coating on a 300M steel substrate. However, it is commonly believed that the tempering process using a conventional furnace is more suitable to achieve a homogeneous microstructure and properties for large-scale components [8].

To the best of our knowledge, there was little reported on the low-temperature tempering heat treatment (<250 °C) applied to laser-cladded Fe-based coatings to date. The present work has developed a novel technique to combine laser cladding with a low-temperature (210 °C) tempering treatment for achieving high-performance AISI 420 SS coatings. The microstructure, mechanical properties and corrosion resistance have been carefully investigated for the pre- and post-tempered specimens, which exhibit superior overall properties to the laser-cladded Fe-based SS coatings previously reported. This study may be of great significance in industrial applications, especially for those requiring a good combination of mechanical properties and corrosion resistance.

## 2. Materials and Methods

### 2.1. Materials

An A36 steel plate with dimensions of 110 mm × 45 mm × 15 mm was used as the substrate material, which was sandblasted and cleaned by acetone prior to the laser cladding process. Spherical gas-atomized AISI 420 SS powder was supplied by Changsha Tianjiu Materials Ltd. (Changsha, China), with a mean particle diameter of 75 μm. The chemical composition of the powder was analyzed by inductively coupled plasma atomic emission spectroscopy (ICP-AES), as listed in Table 1.

**Table 1.** Chemical composition of the AISI 420 SS Powder (wt.%).

| Element | Chemical Composition (wt.%) |
|---------|------------------------------|
| C | 0.26 |
| Cr | 13.15 |
| Ni | 0.55 |
| Mn | 1.02 |
| Si | 0.97 |
| Fe | Bal. |

### 2.2. Coating Preparation and Heat Treatment

A TJ-HL-T5000 5 kW $CO_2$ laser (Wuhan Unity Laser Co., Ltd., Wuhan, China) with coaxial powder feeding and a water cooling system was used to perform the multi-layer laser cladding, as sketched in Figure 1a. The main processing parameters were as follows: Laser power 2.5 kW, laser scanning speed 6 mm·s$^{-1}$, overlapping rate 50% and powder delivery rate 6.5 g·min$^{-1}$. Pure argon was used as both a shielding gas and a carrier gas with a flow rate of 10 L·min$^{-1}$. A 90 mm × 35 mm × 4 mm clad layer was built (Figure 1b), which is referred to as "as-cladded" in this study. The as-cladded specimen was immediately heat treated in a furnace at 210 °C for 1 h, as a case study, followed by cooling in air to room temperature. The latter specimen is referred to as "as-tempered" correspondingly.

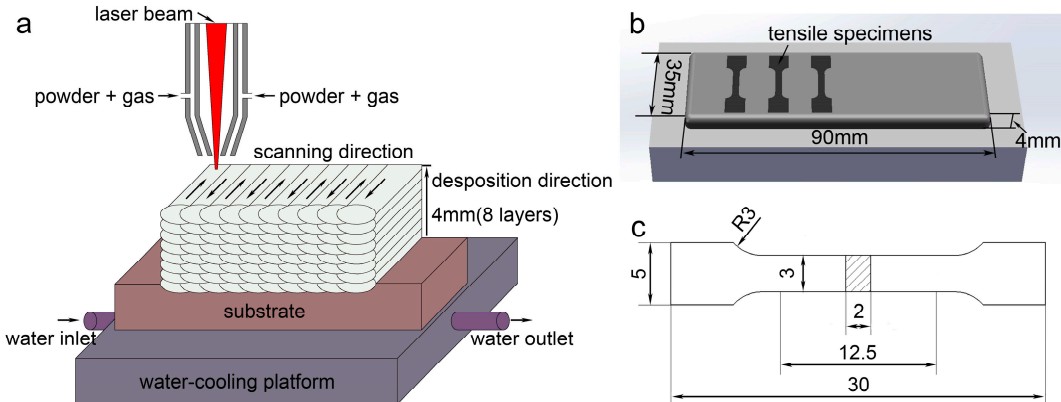

**Figure 1.** (**a**) Schematic of laser cladding process, (**b**) tensile specimens machined from the laser-cladded coating, and (**c**) geometry of the tensile specimens in mm scale.

## 2.3. Characterization

To achieve a flat surface, the excess top ~1 mm of the laser-cladded coating was firstly machined off using a computerized numerical control (CNC) mill. Then, the experimental specimens were extracted from the as-cladded coating and mechanically polished. The phase constituents and microstructure of the as-cladded and as-tempered specimens were carefully examined using a Miniflex600 X-ray diffractometer (XRD, Rigaku Co., Tokyo, Japan) with Cr Kα radiation, a TM3030 scanning electron microscope (SEM, Hitachi Ltd., Tokyo, Japan), and a JEOL-2100 transmission electron microscope (TEM, JEOL Ltd., Tokyo, Japan) operated at 200 kV.

The micro-hardness profile was measured along the cross-sections of specimens using a HMV-2T micro-hardness tester (Shimadzu Co., Kyoto, Japan) with a load of 200 g for 10 s. The room-temperature tensile properties were tested using a PWS-E100 universal testing machine with a cross-head speed of 0.2 mm·min$^{-1}$. Three tensile specimens were extracted from the as-cladded coating of the pre- and post-tempered specimens to examine the repeatability of the results (Figure 1b). The geometry of the tensile specimens according to ASTM E8 [9] is shown in Figure 1c. In order to evaluate the effect of the low-temperature tempering on the corrosion resistance of the as-cladded AISI 420 SS coating, potentiodynamic polarization tests were conducted in a 3.5% NaCl solution at room temperature using a cell with a three-electrode system (CS300, Wuhan Questt Asia Technology Co., Ltd., Wuhan, China). The reference electrode was a saturated calomel electrode (SCE), while the Pt electrode and the specimen were used as the counter electrode and working electrode, respectively. A commercial bulk AISI 420 SS was tested as a reference under the same conditions.

## 3. Results

### 3.1. Phase Analysis

Schaeffler diagrams have been widely utilized to predict the solidification mode of laser cladding metals [10]. The chemical composition of the alloying elements excluding Fe can be simplified by using chromium equivalent ($Cr_{eq}$) and nickle equivalent ($Ni_{eq}$), known as Schaeffler equivalent formulas [10]:

$$Cr_{eq} = Cr + 1.5 \times Si + 0.5 \times Nb \tag{1}$$

$$Ni_{eq} = Ni + 30 \times C + 0.5 \times Mn \tag{2}$$

where each chemical element symbol represents the corresponding weight percentage content. Due to the significant difference in the amount of C, Cr and Ni between the substrate and coating, the $Cr_{eq}$ and $Ni_{eq}$ values were approximately the same with and without, considering the substrate dilution during the laser cladding process. By calculating the formulas of Equations (1) and (2), the $Cr_{eq}$ and $Ni_{eq}$ values were 14.6 and 8.87, respectively, as indicated by the red dot in Figure 2a. It is worth mentioning

that the 0.55 wt % Ni present in this work, as indicated in Table 1, is higher than those reported in AISI 420 SS with a lean Ni element [2–7]. In view of the high hardenability of AISI 420 stainless steel, together with the presence of the austenite stabilizer Ni, it can be reasonably speculated that martensite (M) and austenite (A) coexist in the as-cladded coating in this work.

Figure 2b compares the XRD patterns of the pre- and post-tempered specimens. Apparently, both as-cladded and as-tempered specimens consisted mainly of martensite, austenite, and carbides, such as $Fe_3C$. Similar phases have also been observed in laser-deposited martensitic SS coatings [2,6]. The absence of the $M_{23}C_6$ (*M* represents Cr, Fe etc.) phase (in Figure 2b), however, may be attributed to the small volume fraction confirmed by the subsequent TEM beyond the detection limit of XRD. Notably, the intensity ratio of the peak at 26° to the peak at 45° increased after tempering, suggesting an increase in the volume fraction of carbides in the as-tempered specimen.

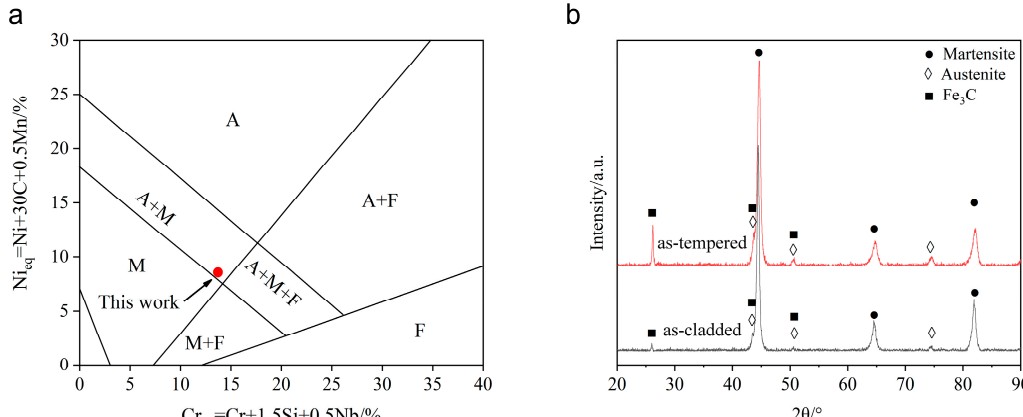

**Figure 2.** (**a**) Schaeffler diagram of AISI 420 SS powder used in this work and (**b**) XRD patterns of the specimens.

## 3.2. Microstructure Characterization

Figure 3 shows the cross-section SEM micrographs of the pre- and post-tempered specimens. It can be clearly seen from Figure 3a,c that the laser-cladded AISI 420 coatings were free of defects such as pores and cracks, and exhibited a good metallurgical bonding between the coating and the substrate before and after tempering. By further comparing the coating zones with high magnifications, the lath-shaped martensite in the as-tempered specimen (Figure 3d) was much finer than that in the as-cladded specimen (Figure 3b). Given that the martensitic transformation start temperature (Ms) of AISI 420 SS is about 160 °C [7], the partitioning of martensite can be expectedly initiated at the tempering temperature of 210 °C used in this work.

Careful TEM characterization was performed to further confirm the phase identification and determine the distribution of each phase. For the as-cladded AISI 420 SS coating, the low-magnification bright-field (BF) image and the selected-area electron diffraction (SAED) pattern indicates that the lath-shaped martensite exists as the matrix on which austenite and secondary particles were randomly dispersed (Figure 4a). The austenite exhibited a darker color due to the different Bragg condition, similar to the TEM observation of the heat-treated Ferrium S53 SS [11].

Figure 4b corresponds to the dark-filed (DF) image and SAED pattern of austenite as indicated in Figure 4a. A dispersion of coarse blocky retained austenite, with a size of about 100 nm in width and 500 nm in length, were discernable in the as-cladded coating. It is well established that the solidification sequence of the AISI 420 SS is as follows: L → L + δ → δ + A → A → M [4]. The fast cooling rate of the laser cladding process and the addition of austenite-promoting element Ni, however, lead to insufficient transformation from A (austenite) to M (martensite). Consequently, martensite and austenite were formed at room temperature.

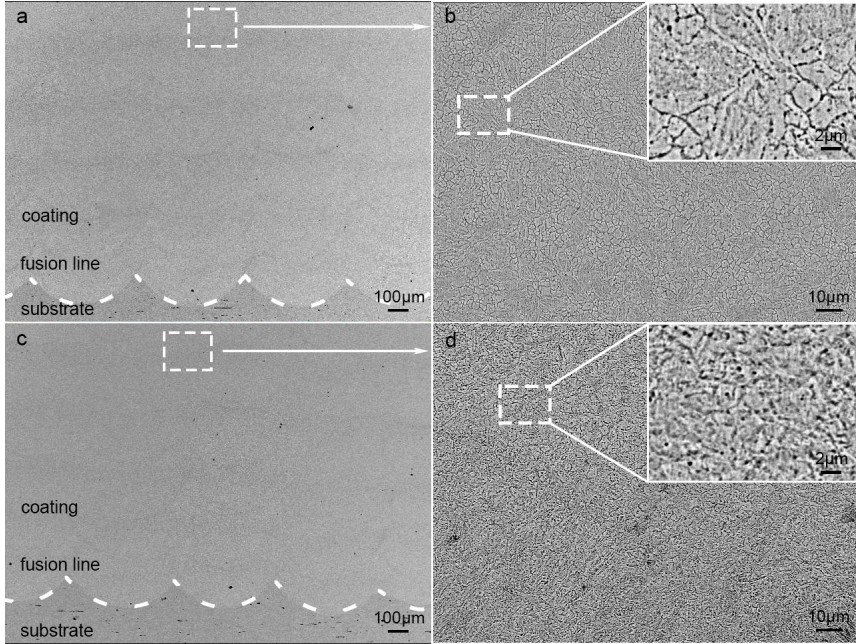

**Figure 3.** The cross-section SEM micrographs of the specimens: (**a**,**b**) as-cladded; (**c**,**d**) as-tempered.

Notably, two types of carbides with different sizes co-existed in the laser-cladded coating (Figure 4c). The majority of the carbide particles were in a dispersive distribution with the size of less than 10 nm, as shown in Figure 4d. These particles can be designated as $Fe_3C$, which is demonstrated by the SAED pattern inset of Figure 4d. On the contrary, the others with a larger size of 50 nm in diameter can be indexed as the $M_{23}C_6$ phase (Figure 4e), evidenced by the SAED pattern inset of Figure 4e. In addition, a large amount of dislocation could be observed in the as-cladded specimen (Figure 4f). This was caused by the rapid heat dissipation and the resultant high stresses during the laser cladding process, consistent with previous report [12]. The high dislocation density is a vital factor impeding the lattice movement, which is expectably beneficial for strengthening of the as-cladded coating [8].

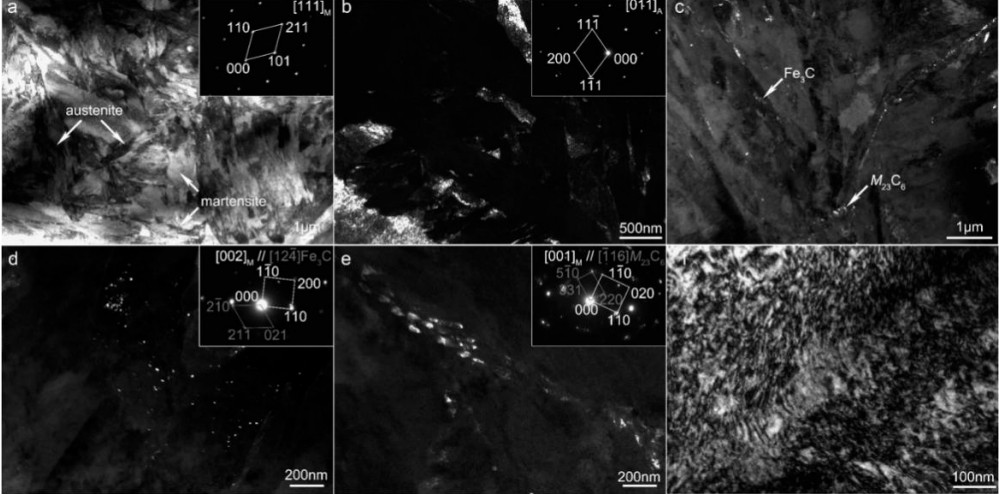

**Figure 4.** TEM analysis of the as-cladded specimens: (**a**) BF image; (**b**) DF image and SAED pattern of austenite; (**c**) DF image of carbides; (**d**) DF image and the SAED pattern of $Fe_3C$; (**e**) DF image and the SAED pattern of $M_{23}C_6$; (**f**) dislocation.

Figure 5 shows the TEM analysis of the as-tempered specimen. In comparison with the as-cladded specimen, the microstructure was somewhat refined as seen in Figure 5a, consistent with the SEM observations as shown in Figure 3. There were two types of austenite as marked by arrows after tempering (Figure 5b). The majority of austenite were needle-like, with a width of 30 nm and a length of 50–500 nm, which were distributed on the boundary of martensite laths. This type of need-like reverted austenite usually originates from the partial reversion of martensite during tempering and/or recrystallization process [13]. Notably, the finely reverted austenite was different from the blocky retained austenite and also more stable, which is believed to enhance the toughness and plasticity of the Fe-based alloy without reducing its strength [11]. Additionally, the amount of the retained austenite was reduced due to its decomposition during tempering [14].

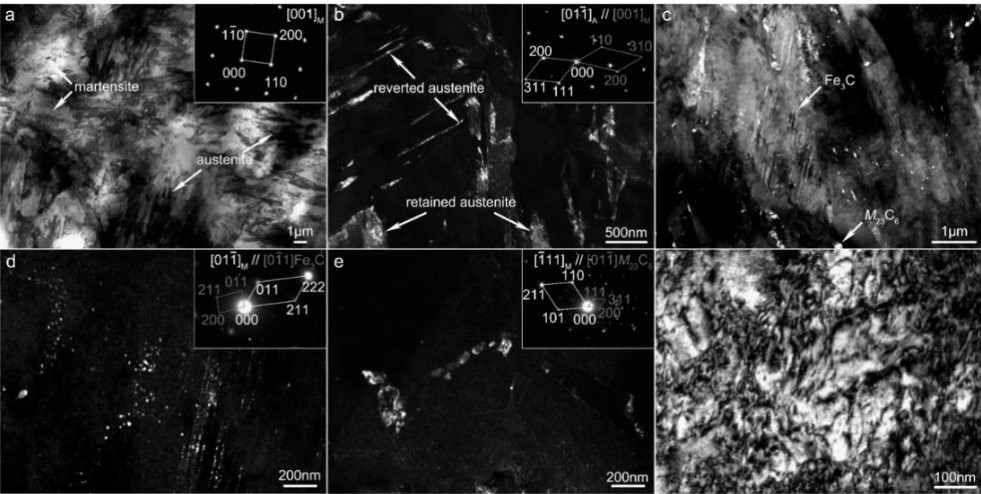

**Figure 5.** TEM analysis of the as-tempered specimens: (**a**) BF image; (**b**) DF image and the SAED pattern of austenite; (**c**) DF image of carbides; (**d**) DF image and the SAED pattern of $Fe_3C$; (**e**) DF image and the SAED pattern of $M_{23}C_6$; (**f**) dislocation.

Similarly, two types of granular carbides can be observed on the martensite matrix (Figure 5c,d,e). It can be noticed that the number density of $Fe_3C$ carbides increased after tempering, which is in good agreement with the formation of nano-sized $M_3C$ carbides after tempering in the temperature range of 150–350 °C, as reported by other studies [15,16]. The increased precipitation of $Fe_3C$ carbides in the as-tempered specimen can be explained by the decomposition of the supersaturated martensite matrix, due to the rapid solidification of the laser cladding process. It is well-known that the element carbon can diffuse readily at 100–200 °C and segregate at lattice defects [3], consequently precipitate in the formation of carbides during the subsequent tempering process. Comparatively, the number density of $M_{23}C_6$ particles almost remained the same, which was demonstrated to precipitate preferably at a higher tempering temperature above 500 °C in previous work [15–18].

High-density dislocation, developed during the laser-cladding process, as illustrated in Figure 4f, can be the driving force for facilitating the nucleation of reverted austenitic grains and carbides in the subsequent tempering treatment [12,19]. The number density of dislocation decreased after a low-temperature tempering (Figure 5f), resulting in less stain hardening.

*3.3. Mechanical Properties*

Figure 6 displays the test results of the mechanical property of the pre- and post-tempered specimens. The micro-hardness profile of the cross-section specimens revealed a consistently uniform hardness distribution across the coatings, as seen in Figure 6a. The average micro-hardness of the as-cladded coating and the as-tempered coating was 488.3 and 455.7 HV, which was 2.9 and 2.7 times the micro-hardness of the baseline material, respectively. The decrease in micro-hardness after tempering

was mainly caused by the partial decomposition of the supersaturated lath-shaped martensite, and also the decrease of the dislocation density.

Figure 6b shows the typical tensile curves of the pre- and post-tempered specimens, and the tensile properties are summarized in Table 2. Significantly, the as-tempered specimen exhibited striking tensile properties: UTS (ultimate tensile strength) of 1690 MPa, $YS_{0.2}$ (0.2% yield strength) of 1109 MPa and elongation of 15.8%, respectively. Compared with the as-cladded specimen, the ductility was enhanced by 2.3 times, while there was only a 6.2% reduction in the $YS_{0.2}$ and the UTS after a low-temperature tempering treatment at 210 °C. This is in good agreement with the report that the ductility of martensitic SS can be significantly enhanced by tempering the martensite even at low temperature (200–300 °C) [7]. The reasons for the high ductility of the as-tempered specimen could be associated with the following factors: (1) The refined microstructure after tempering; (2) the occurrence of the tempered martensite and reverted austenite offering a higher ductility and toughness [11,19]; (3) the deduced dislocation density after tempering; and (4) the deduced residual stress produced during the laser cladding process [6].

To make it clear, Table 2 lists the data of the UTS, $YS_{0.2}$ and elongation of AISI 420 SS coating developed in this study, and is compared with other Fe-based SS coatings from the literature, prepared by the laser cladding technique [10,12,18–20]. It can be easily concluded that the as-tempered AISI 420 SS coating in this work exhibited a striking combination of high strength and high ductility.

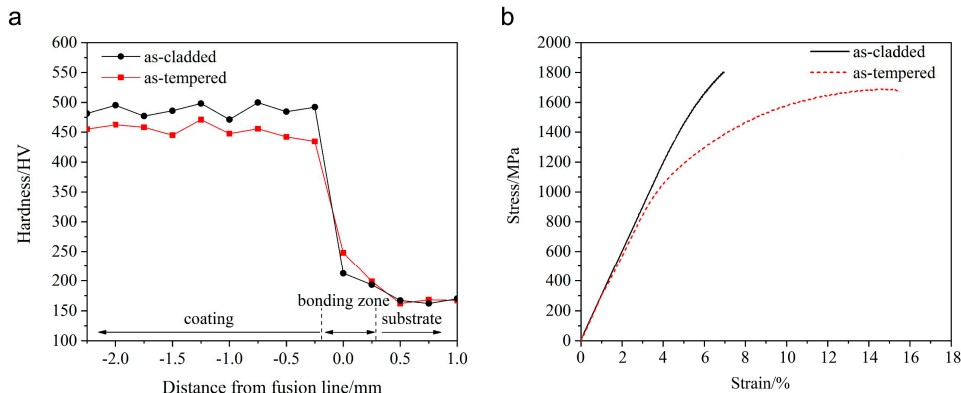

**Figure 6.** Mechanical properties of the specimens: (**a**) micro-hardness distribution profiles of the cross-section specimens; (**b**) tensile curve of coatings.

**Table 2.** Comparison of the Tensile Properties from different laser cladded Fe-based SS coatings.

| Coatings | Treatment States | UTS (MPa) | $YS_{0.2}$ (MPa) | Elongation (%) | Ref. |
|---|---|---|---|---|---|
| AISI 420 | as-cladded | 1802 | 1306 | 6.9 | This work |
| AISI 420 | as-tempered (210 °C, 1 h) | 1690 | 1109 | 15.8 | This work |
| AISI 431 | as-tempered (680 °C, 2 h) | 905 ± 6 | – | 16.3 ± 0.8 | [18] |
| AISI 431 | as-heat treated (1050 °C, 45 min, oil-quenching + 315 °C, 3 h, tempering) | 1283 ± 16 | – | 14.5 ± 1.5 | [18] |
| SAF2507 | as-cladded | 1321 ± 48 | 1214 ± 43 | 8.6 ± 1 | [19] |
| AISI 308L | as-cladded | 548 | 377 | 40 | [20] |
| UNS S31803 | as-cladded | 940 | 890 | 12 | [12] |
| UNS S31803 | as-annealed (1000 °C, 5 min) | 770 | 600 | 28 | [12] |

Figure 7 illustrates the fracture morphology of the tensile specimens at different magnifications. The as-cladded specimen exhibited a typically brittle fracture, characterized by the flat fracture surface with cleavage steps, as shown in Figure 7a,b. Comparatively, the fracture morphology of the as-tempered specimen was obviously different from that of the as-cladded specimen, which consisted of massive dimples and a few tearing ridges (Figure 7c,d). The small ductile dimples might be attributed to the fined and uniform microstructure, while the tearing ridges were probably caused by the presence of carbides such as $M_{23}C_6$ and $Fe_3C$. Some micro-cracks could be observed between

the carbide particles, leading to the quasi-cleavage fracture characteristics as indicated in Figure 7d. The fractography is therefore consistent with the measured tensile mechanical properties as listed in Table 2.

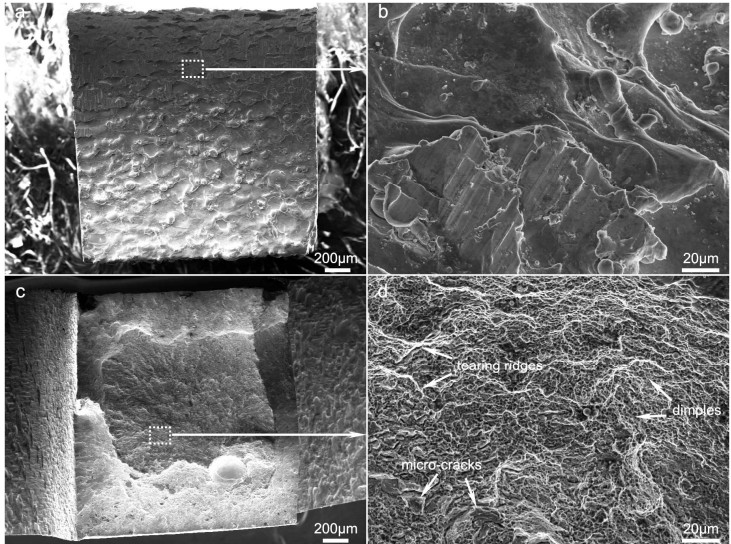

**Figure 7.** Fractography of tensile specimens at different magnifications: (**a**,**b**) as-cladded specimen; (**c**,**d**) as-tempered specimen.

## 3.4. Electrochemical Properties

Figure 8 displays the polarization curves of the specimens immersed in 3.5 wt.% NaCl solution, and the corresponding electrochemical parameters are summarized in Table 3. Apparently, both pre- and post-tempered AISI 420 SS coatings showed a comparable corrosion resistance with the commercial bulk AISI 420 SS. The polarization curve of the as-tempered coating shifted to the right (higher current density) and downward (less noble corrosion potential), compared with that of the as-cladded coating. This indicates that the low-temperature tempering treatment lead to a slight decrease in corrosion resistance of the laser-cladded coating. The higher corrosion rate of the as-tempered specimen can be explained by the Cr depletion theory [21]. On one hand, the segregated Cr-rich carbides make the immediate Cr-depleted vicinity less passive and act as the initial sites for pitting corrosion. On the other hand, the precipitation of Cr-rich carbides partially consumes the element Cr. This weakens the formation of passive $Cr_2O_3$ film preventing the Fe-based SS coating from the corrosion. Actually, the effect of tempering on the corrosion resistance of the Fe-based SS microstructure is consistent with the literature [13].

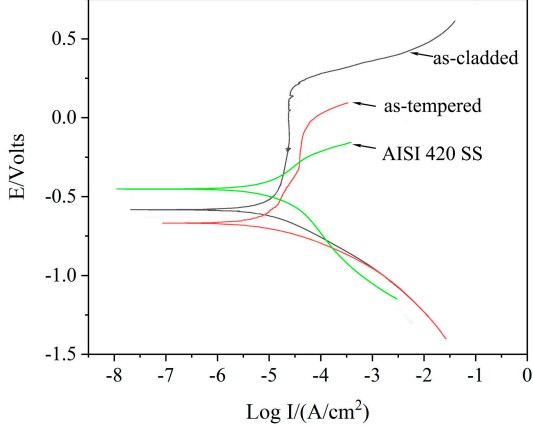

**Figure 8.** The polarization curves of the pre- and post-tempered specimens.

**Table 3.** Electrochemical parameters of the pre- and post-tempered specimens.

| Specimens | $E_{corr}$ (V) | $I_{corr}$ (A·cm$^{-2}$) | Corrosion Rate (mm·a$^{-1}$) |
|---|---|---|---|
| AISI 420 SS as-cladded coating | −0.5813 | $3.838 \times 10^{-6}$ | 0.03723 |
| AISI 420 SS as-tempered coating | −0.6671 | $4.271 \times 10^{-6}$ | 0.04194 |
| Bulk AISI 420 SS | −0.4368 | $3.630 \times 10^{-6}$ | 0.03066 |

## 4. Conclusions

In this study, the effects of low-temperature tempering (210 °C) on the microstructure and properties of the laser-cladded 420 SS coating have been carefully investigated. Conclusions can be drawn as follows.

- The phase constituents of both the pre- and post- laser-cladded AISI 420 SS coatings were mainly martensite, austenite, nano-sized Fe$_3$C and $M_{23}C_6$ carbides. After tempering treatment, the martensite was refined and the dislocation density was decreased, along with the massive formation of the reverse austenite and nano-sized Fe$_3$C carbides.
- The low-temperature tempering treatment caused a slight decrease (6%–10%) in micro-hardness and strength, but a significant increase (1.6 times) in the ductility of the laser-cladded coating. The as-tempered AISI 420 SS coating exhibited striking mechanical properties with a UTS of 1690 MPa, YS$_{0.2}$ of 1109 MPa and elongation of 15.8%, superior to the laser-cladded Fe-based SS coatings reported in the literature.
- Both the pre- and post-tempered specimens exhibited a similar corrosion resistance with the bulk AISI 420 SS. However, the corrosion resistance was slightly decreased after tempering.

**Author Contributions:** Conceptualization, H.Z.; Methodology, H.Z. and C.Q.; Software, Y.L.; Validation, Y.L., B.L. and Z.Z.; Formal Analysis, H.Z., Y.L. and B.L.; Investigation, Y.L.; Resources, C.Q.; Data Curation, B.L. and Z.Z.; Writing—Original Draft Preparation, H.Z.; Writing-Review & Editing, H.Z.; Visualization, Y.L., B.L. and Z.Z.; Supervision, C.Q.; Project Administration, C.Q.; Funding Acquisition, C.Q.

**Funding:** This work was funded by National Natural Science Foundation of China (No. 51474130) and National Key R&D Program of China (No. 2018YFB1105803).

**Conflicts of Interest:** The authors declare no conflict of interest.

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
