# Peer review of "Effects of Low-Temperature Tempering on Microstructure and Properties of the Laser-Cladded AISI 420 Martensitic Stainless Steel Coating"

_coatings, doi:10.3390/coatings8120451_

Reviewer 1 Report

This paper deals with the well-known martensitic stainless steel AISI 420, which is strengthened by heat treatment and has high wear resistance. Martensitic-grade steel is widely used in industry: springs performing at temperatures up to 400–450, ball bearings, cutting and measuring tools. In this paper, as usual for the laser powder cladding processes, an alloy of maximum strength and hardness is formed, its parameters quite similar to the parameters of industrial rolled products with UTS (ultimate tensile strength) σB = 1700 MPa and relative elongation of δ = 5.5%. (see for example http://metallicheckiy-portal.ru/marki_metallov/stn/AISI420).

It is well known that the parameters of AISI 420 steel can be varied within a wide range by heat treatment methods. In this work, the effect of low-temperature thermal treatment at 210° C is investigated. The microstructure, mechanical properties and corrosion resistance were carefully investigated for the pre-and post-thermal treatment.

Specific comments.

1. The thermal treatment term, namely the “tempering process” is incorrectly used in the work. The heat treatment process used in the work (“The as-cladded specimen was heat treated in the furnace at 210 °C for 1 hour, followed by cooling in air to the room temperature») means the annealing process.

2. In the paper, there is no information on the beam motion pattern and the number of layers when creating the clad layer with dimensions of 90 mm × 35 mm × 4 mm.

3. Figure 3. «The cross-section of SEM micrographs of the specimens. (a, b) as-cladded; (c, d) as-tempered» yields very little information and should be removed.  A dendritic structure is formed in the laser-cladded AISI 420 coatings, which is missed in this figure. It means that the cross section of the layer is not well prepared.  (For example, compare the results of similar work [17]. Figure 4. ... (b) SEM micrograph of the solidification structure of the as-deposited steel).

4. It is necessary to specify the experimental variation of the UTS data (MPa), YS0.2 (MPa) and elongation (%) for AISI 420 as-cladded and as-tempered. (Table 2. Comparison of the tensile properties from different laser cladded Fe-based SS coatings).

Author Response

Dear Reviewer:

We are truly grateful to your critical comments and thoughtful suggestions. Based on these comments and suggestions, we have made careful modifications on the original manuscript. Please find the point-by-point respond in the attachment. Thank you!

Reviewer 2 Report

Remarks

1) The XRD patterns are not properly characterized. The problem arises from the usage of the Cu radiation source when conducting the experiments. The Cu Ka radiation is the most widely used when performing XRD. However samples that are rich in Fe, Cr, Mn will fluoresce under the incident Cu Ka beam and will create polychromatic radiation. These can lead to strange shaped and elevated backgrounds (as shown in Figure 2b) and incorrect evaluation of the phases.A chromium X-ray source with a vanadium metal or compound filter to reduce the Kβ radiation is recommended. Chromium radiation produces a minimum of X ray fluorescence of iron and allows for the separation of carbide peaks from austenite and ferrite peaks. The peaks of ferrite and martensite are coincident when the carbon content is relative small.

Proposal to the authors: Either the XRD pattern should be deleted (it gives relatively small new insight since the microstructures are also characterized by TEM and SEM) or the XRD experiments should be performed again using other radiation source (not a Cu one!)

2) The manuscript seems to be a case study for a specific coated material, tempered at one low temperature (e.g., 210oC for 1 hour). This does not give new insight in the scientific community. On the contrary it makes the manuscript look like a technical report. Therefore the authors should justify the choice of this specific heating cycling. Why did the authors choose this heating process? Is it the optimum? Have the authors tested other heating temperatures/durations as well?

3) The novelty of the paper should be written at the introduction section and not at the conclusion section (last paragraph).

4) Some details should be given for the electrochemical tests. Did the author use a three-electrode cell? What was the working and reference electrode? Additionally, the authors should mention that the micro-hardness was measured by the use of Vickers hardness test.

5) In the SEM images (Figure 3) the structures are barely visible. A contrast enhancement should be done especially for fig3 a and c.

6) The author use several grade systems (Chinese grade and AISI grade) for steels throughout the manuscript which is rather confusing (e.g., AISI 420 SS, 1Cr13 SS, Q235). They should keep a uniform preferably the AISI/SAE steel grades system.

7) Why did the authors used a 1Cr13 SS (which is the AISI 410 SS) as a reference material for the potentiodynamic tests?

8) Is the measured potential E in Figure 8 measured versus the potential of the saturated calomel electrode (SCE)?

9) The conclusion section is too large. The author should mention only the major conclusions of the manuscript.

10) For reasons of clarity the results section should be divided in subsections. 

Author Response

(The authors gave the same response as above.)

Reviewer 3 Report

(1) L87 “ the laser-cladded coating was firstly machined off using a CNC mill. Then, three tensile specimens were extracted from the as-claded coating of the pre- and post-tempered specimens to examine the repeatability of the results (Figure 1b).

It seems that there were three tensile specimens to examine the repeatability of the results. However, Figure 6 (b) shows only one result for each treatment of as-cladded and as-tempered. Please answer this question?

(2) In Figure 2 (b), XRD patterns were shown. The XRD tested surface were polished? The surface of cladding layer has a possibility to be oxidized, even though Ar gas shielding. If the cladding samples were polished before XRD testing, author should describe as the sample preparation procedure.

(3) The cladding sample thickness was 4 mm. Meanwhile, the position of Figure 3 of SEM micrographs in the cladding layer were sited at 600 µm from the substrate. Why author selected the observation point at near substrate, not center of the cladding layer.

In addition, was the cooling rate in the cladding layer changed with the distance from the substrate? Because the substrate was cooled by water flow, then it is considered that the cooling rate at near substrate was faster than the surface of cladding layer. Did the microstructure of cladding layer change with the distance from the substrate? 

Author Response

Dear Reviewer:

We are truly grateful to your critical comments and thoughtful suggestions. Based on these comments and suggestions, we have made careful modifications on the original manuscript. Please find the point-by-point respond in the attachment. Thank you!

Round  2

Reviewer 2 Report

The authors have successfully managed to answer all the remarks poposed by this reviewer.  This reviewer believes that the paper is now suitable for publication.

Reviewer 3 Report

This paper was modified in accordance with the review’s suggestions fully.

Therefore, this paper should be accepted.